# Prevalence and Characterization of Salmonella during Pork Sausage Manufacturing

**DOI:** 10.3390/microorganisms12081599

**Published:** 2024-08-06

**Authors:** Lauren R. Wottlin, Roger B. Harvey, Keri N. Norman, Robert E. Droleskey, Kathleen Andrews, Steve J. Jackson, Robin C. Anderson, Toni L. Poole

**Affiliations:** 1Food and Feed Safety Research Unit, Agricultural Research Service, U.S. Department of Agriculture, 2881 F&B Road, College Station, TX 77845, USA; laurenrwottlin@gmail.com (L.R.W.); roger.harvey@usda.gov (R.B.H.); bob.droleskey@usda.gov (R.E.D.); kate.andrews@usda.gov (K.A.); robin.anderson@usda.gov (R.C.A.); 2Department of Veterinary Integrative Biosciences, Texas A&M University, 3201 Russell Long Boulevard, College Station, TX 77843, USA; knorman@cvm.tamu.edu; 3Department of Veterinary Physiology and Pharmacology, Texas A&M University, College Station, TX 77843, USA; steve.jackson@siu.edu

**Keywords:** pork, sausage, Salmonella, lymph nodes, Enterobacteriaceae, antimicrobial resistance

## Abstract

Pork carcasses and meat may harbor Salmonella and may contaminate other products during harvest and fabrication. Sources of contamination include manure on hides, environmental contamination, ingredients from external sources, and lymph nodes. Swine lymph nodes are often incorporated into ground meat, as their anatomical location makes removal labor prohibitive. A sausage processing plant in the midwestern United States was sampled monthly (except for December) from May 2021 to April 2022 to enumerate Salmonella and Enterobacteriaceae (EB) throughout the sausage manufacturing process to determine high-risk stages and efficiency of existing in-plant interventions. Salmonella serotypes and antimicrobial susceptibility were evaluated on a subset of isolates recovered at the end phases of sausage production. In each collection, samples were taken from the carcasses of eight sows through 11 stages of sausage manufacturing. A total of 830 samples were cultured. Thirty-four Salmonella were isolated from the final three production stages; of these, there were eleven serotypes. Three isolates displayed resistance to ampicillin, whereas the remainder of the isolates were pan-susceptible to the antimicrobials tested. Salmonella and EB were significantly reduced (*p* < 0.001) by acid washes at different stages of production, and the results point to the beneficial effects of interventions to lessen Salmonella concentrations in retail products.

## 1. Introduction

The World Health Organization (WHO) has established the Foodborne Disease Burden Epidemiology Reference Group (FERG), and they have found that foodborne diseases are comparable to other infectious diseases in terms of estimates of incidence, mortality, and disease burden [1]. Non-typhoidal Salmonella *enterica* subspecies *enterica* (hereafter called Salmonella) is the leading cause of foodborne illness in the United States and is estimated to cause 1.3 million illnesses and more than 23,000 hospitalizations and 450 deaths in humans each year [2]. The financial losses due to Salmonella are estimated by the United States Department of Agriculture’s Economic Research Service (ERS) to be approximately 4.1 billion annually [3].

Pork is the second most consumed meat globally [4], and the National Pork Board estimates that 25% of U.S. pork, more than 2.2 million metric tons of pork and pork-related products, are exported annually [5]. Many governments have established epidemiology programs to monitor Salmonella in pork and instituted programs for mitigation [6]. Swine are a natural host for Salmonella, and infections are often asymptomatic and self-limiting, making eradication difficult [6]. Infection occurs via the fecal–oral route [7]. It is believed that within 24 h of initial infection, Salmonella may be able to establish a persistent infection in mesenteric lymph nodes [8,9]. Swine lymph nodes are often incorporated into ground meat, as their anatomical location makes removal labor prohibitive. As lymph nodes have been shown to exhibit substantial Salmonella prevalence in healthy cattle and swine at slaughter [10,11], the inclusion of lymph nodes in edible products may present a food safety threat. Culled breeding sows are typically used exclusively for sausage production, suggesting most of the lean tissue is ground and likely contains some lymph nodes. Additionally, other sources of Salmonella (e.g., manure on hide, environmental contamination, ingredients from external sources, etc.) may contaminate meat during processing [12,13,14].

Sample collection around each control point in the harvest and fabrication process is necessary to evaluate the efficacy of in-plant interventions and to monitor any changes in microbial resilience and adaptation. Sensitivity analyses have indicated that in-plant interventions are more impactful with a lower cost–benefit ratio than on-farm interventions [13]; therefore, it is important to characterize and monitor the burden of Salmonella and other microbes which persist to the end of the manufacturing process.

The objectives of this study were to enumerate Salmonella and Enterobacteriaceae (EB) cultured from samples collected throughout the sausage manufacturing process to determine high-risk stages and the efficacy of existing in-plant interventions. Furthermore, Salmonella serotypes and antimicrobial susceptibility were evaluated on a subset of isolates recovered at the end phases of sausage production.

## 2. Materials and Methods

### 2.1. Sample Collection

Samples from sows were collected monthly from May 2021 to April 2022, except December, from a single pork sausage facility in the midwestern United States. Farms supplying sows to the facility varied. Production stages represented were as follows: (1) hide swab, pre-NaOH wash; (2) hide swab, post-NaOH wash; (3) carcass swab, pre-evisceration; (4) carcass swab, post-split and trim; (5) carcass swab, post-lactic acid wash (6) head swab, pre-wash; (7) head meat, post-peracetic acid and lactic acid washes; (8) coarsely ground head and carcass meat from all 8 sows combined; (9) supplemental pork fat; (10) spiced, chilled, and fat-corrected batter; and (11) finished product samples from retail store-ready packaging (Table 1). For each collection, samples were taken from the same 8 sows. Samples 1–7 are from each of eight individual sows per collection, while stages 8, 10, and 11 involved two samples, each collected at the beginning, middle, and end of that production stage, for a total of six per stage. Stage 9 came from external sources. Carcasses (ham, back, belly, jowl area) and heads were swabbed over approximately 100 cm^2^ with sterile pre-moistened sponges (Sponge Stick, 3M Corp., St. Paul, MN, USA) according to manufacturer’s recommendations. Each sponge was placed in a sterile collection bag with 10 mL of buffered peptone water. Meat samples (approximately 500 g each) were collected and then placed in sterile collection bags. Personnel from the manufacturing plant collected all samples and provided data on source farms. All samples were transported overnight on ice to the USDA–ARS laboratory in College Station, TX, USA.

### 2.2. Sample Processing

All media were sourced from Becton, Dickinson and Company (Sparks, MD, USA) unless otherwise specified. Each sample was enumerated for EB and Salmonella. Hide swab samples (production stages 1 and 2) were enumerated using a spiral plater and cultured on xylose lysine desoxycholate (XLD) agar medium (Remel, St. Louis, MO, USA) with 4.6 mL/L Tergitol, 15 mg/L novobiocin, and 5 mg/L cefesulodin (XLDtnc, Sigma, St. Louis, MO, USA) for Salmonella detection and MacConkey controlled swarming (MacCS) media for EB detection. Salmonella was enumerated after the XLDtnc plates were incubated at 37 °C for 18–22 h and then at room temperature for 18–22 h [15]. The MacCS plates were incubated at 37 °C for 18–22 h then EB colonies were counted. Samples from production stages 3–8 were quantified by applying 1 mL aliquots directly from carcass swab bags or from meat samples (meat samples with 80 mL tryptic soy broth [TSB] added) to Petrifilm EB count plates (3M Food Safety, St. Paul, MN, USA) in duplicate and incubated at 37 °C for 18–22 h. Next, each Petrifilm was transferred onto XLDtnc for Salmonella enumeration and MacCS for EB enumeration and incubated for 18–22 h at 37 °C. Black colonies suspected to be Salmonella on XLDtnc were confirmed as Salmonella by picking one representative colony per plate and inoculating to triple sugar iron and lysine iron slant agars, with both slants turning black and possibly producing a bottom gas pocket.

For Salmonella prevalence analysis, 80 mL of TSB was added to swab samples, and 315 mL of TSB was added to meat samples. All samples were incubated at room temperature for 2 h, then 42 °C for 6 h (swab samples) or 12 h (meat samples) and held at 4 °C overnight until processed the next day. Next, a 1 mL aliquot from each enrichment sample was removed and mixed with 20 µL of anti-Salmonella beads (IMS) and subjected to anti-Salmonella immunomagnetic separation as previously described [12]. Beads were added to 3 mL of Rappaport Vassiliadis (RV) and incubated at 42 °C for 18–22 h. The RV enrichment was then swabbed onto Brilliant Green Agar with sulfadiazine (BGAs; 80 mg/L) and incubated at 37 °C for 18–22 h. Up to three pink suspect Salmonella isolates were picked for confirmation on triple sugar iron and lysine iron slant agars. Confirmed Salmonella isolates were inoculated to cryovials and stored at −80 °C until genomic and antimicrobial susceptibility testing occurred. Post-enrichment prevalence of EB was not determined or recorded.

### 2.3. Antimicrobial Susceptibility Testing

One isolate per positive sample recovered from stages 8–11 (post-wash meat products) was selected for antimicrobial susceptibility analysis (*n* = 34). Isolates from pre-wash samples were not tested because they did not enter the food chain. Susceptibility to 14 antimicrobial agents was determined by use of an automated micro-broth dilution method (Sensititre Gram Negative NARMS Plates CMV3AGNF, TREK Diagnostics Inc., Oakwood Village, OH, USA) according to the manufacturer’s recommendations. Susceptible, intermediate, or resistant classification was determined using breakpoints established by the Clinical and Laboratory Standards Institute [16] or by National Antimicrobial Resistance Monitoring System breakpoints [17] when CLSI criteria were not established.

### 2.4. Serotyping

This same subset of isolates was sent to the USDA–APHIS National Veterinary Services Laboratory (Ames, IA, USA) for traditional serotyping using polyvalent and single-factor antisera or molecular typing using a multiplex nucleic acid assay.

### 2.5. Statistical Analysis

All data were analyzed with JMP 15 software (SAS Institute Inc., Cary, NC, USA). Mixed models included sample type as a fixed effect, and month (representing supplier) was used as a random effect in the model. Prevalence data were analyzed using nominal logistic regression, with chi-square tests for significance. For quantitative analysis of Salmonella, samples that were below the limit of detection during initial enumeration but were positive following enrichment were assigned a standardized concentration below the limit of detection determined for each sample type (stages 1 and 2, 10 CFU/mL; stages 3–5, 2 CFU/mL; stage 6, 4 CFU/mL; stages 7–11, 1 CFU/mL). A similar methodology of concentration correction could not be followed for EB as enrichment was not completed for EB samples; thus, the reader should consider that some of the “zeroes” included were likely below the limit of detection. Quantitative concentration data were log-transformed prior to analysis. Concentration data were analyzed using linear mixed effect regression models, with Student’s *t* test for pairwise comparisons of treatment means when warranted. Significance was declared at *p* ≤ 0.05.

## 3. Results

A total of 830 samples were collected for this study. Samples on four collection months were sourced from domestic suppliers, while the remaining seven months were sourced from Canadian suppliers. Neither the supplier nor the farm within the supplier were replicated across any two months.

The concentration and pre-enrichment prevalence of EB can be found in Table 2. Concentrations of EB on individual samples ranged from presumed 0 to 7.53 log_10_ CFU/mL. It should be noted that post-enrichment analyses were not performed for EB; therefore, samples identified as “zero” may have been below the limit of detection. The greatest EB concentrations were found in the first five stages of the manufacturing process. Nevertheless, the absolute prevalence of EB persisted to the final product, though concentrations at that phase were very low (Table 2).

Table 3 presents the mean Salmonella concentrations of each sample type. Similar to EB results, Salmonella concentrations of individual samples ranged from 0 to 6.63 log_10_ CFU/mL, with the greatest concentrations also found on pre-wash hide swabs. Concentration was very low in the latter half of the stages of the manufacturing process. The prevalence of Salmonella in the pre-enriched final product was 2%, whereas Salmonella prevalence in the post-enrichment final product was 15% (Table 3).

Figure 1 graphically displays the prevalence of Salmonella at the different stages of production. Both pre-enrichment and post-enrichment show a similar trend of decreasing Salmonella prevalence throughout production.

Salmonella isolates that originated from coarsely ground head and carcass meat, blended batter, and finished product sources were further evaluated for serotype and antimicrobial susceptibility (Appendix A). There were 34 isolates recovered that represented 11 different serotypes, the most frequent being Eko (*n* = 8), Anatum (*n* = 5), Johannesburg (*n* = 4), London (*n* = 4), and Senftenberg (*n* = 4). Antimicrobial susceptibility testing indicated that three isolates displayed resistance to ampicillin alone, whereas the remaining isolates were pan-susceptible to the antimicrobials tested.

## 4. Discussion

Pork sausage has been implicated in numerous human salmonellosis outbreaks worldwide [18,19,20]. Salmonella has been recovered from pork sausage for many decades, and the same serotypes, including Derby and Anatum, have consistently recovered [14,21,22,23]. Mattick et al. [24] reported that 8.6% of fresh and frozen sausage samples in Devon, England, were Salmonella positive. Broughton et al. [25] reported 1.7% and 4.4% Salmonella prevalences in Irish retail pork sausages during two different sampling periods. Mamber et al. [23] reported a Salmonella prevalence of 0.05% (8/14,800) over an eight-year period in the United States. The present study reported 2% (1/66) pre-enrichment prevalence and 15% (10/66) post-enrichment prevalence in the final product, which seems consistent with the previous studies.

In ready-to-eat meat and poultry products sampled from 2005 to 2012, *S*. Typhimurium and *S*. Infantis were the most common serotypes recovered from positive samples in the United States [23]. In the present study, there were no *S*. Typhimurium and only two *S*. Infantis of the 34 isolates. Rather, the most common serotypes in this study were *S.* Eko and *S.* Anatum.

Salmonella outbreaks in France in 2010 and Spain in 2016 were traced back to the consumption of dried sausage [18,19], whereas the causative factor for a Salmonella outbreak in Denmark in 2018–19 was the consumption of raw pork sausage and pork products [20]. In those studies [18,19,26], most of the pork products were intended to be consumed uncooked, whereas the products of the present study were intended to be cooked before consumption. Consequently, the likelihood of illness from this product may be lower than in those reported above.

Vieira–Pinto et al. [27] reported that Salmonella was detected in 13% of post-evisceration pig carcasses in a plant in northern Portugal. They also reported that 10–19% of tonsils and mandibular and ileocolic lymph nodes carried Salmonella, which represents a potential contamination source when whole or parts of these tissues are included in ground meat. Head meats are known to be associated with a relatively high prevalence of Salmonella [28], possibly due to the inclusion of lymphatic tissues and salivary glands near the oral cavity in head meats [27,29]. Although the design of this study precluded the collection of lymph nodes, including lymph nodes in future studies will help pinpoint higher-risk areas during sausage production.

This study represented an eleven-month limited monitoring system to evaluate the effectiveness of Salmonella intervention strategies. From the data gathered, it is apparent that the application of antibacterial washes/sprays made substantial reductions in Salmonella concentrations from the beginning to the end of the assembly line. These results agree with those reported by others [30,31] in which sulfuric, peracetic, lactic, and citric acids were effective for Salmonella reduction when used as washes on pork head meat. There were variations in outcomes from month to month that could suggest seasonal differences, differences in employee population, differences in the supplying farms, or possibly differences in sample collection because plant personnel collected all samples, but the cause of variation could not be determined. It is clear, however, that interventions used in this plant effectively decreased bacterial numbers in edible products.

## Figures and Tables

**Figure 1 microorganisms-12-01599-f001:**
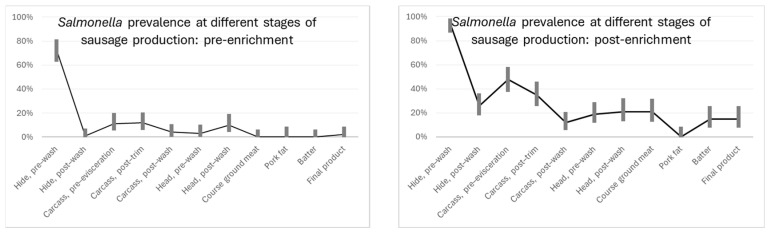
Salmonella prevalence at different stages of sausage production, pre- and post-enrichment.

**Table 1 microorganisms-12-01599-t001:** Production stages where samples were collected.

Production Stage	Sample
1	Hide swab, pre-NaOH wash
2	Hide swab, post-NaOH wash
3	Carcass swab, pre-evisceration
4	Carcass swab, post-split, and trim
5	Carcass swab, post-lactic acid wash
6	Head swab, pre-peracetic acid and lactic acid washes
7	Head meat, post-peracetic acid and lactic acid washes
8	Coarsely ground head and carcass meat from all sows combined
9	Supplemental pork fat
10	Spiced, chilled, and fat-corrected batter
11	Finished product samples from retail store-ready packaging

**Table 2 microorganisms-12-01599-t002:** Pre-enrichment prevalence of Enterobacteriaceae from pork samples (*n* = 830) collected at various stages of sausage manufacturing.

Stage	Samples	Positives	Prevalence	Log_10_ CFU/mL
Hide, pre-wash	88	88	100%	6.13 ^a^
Hide, post-wash	88	11	13%	0.43 ^efg^
Carcass, pre-evisceration	88	74	84%	1.74 ^b^
Carcass, post-trim	85	78	92%	1.93 ^b^
Carcass, post-wash	84	12	14%	0.22 ^g^
Head, pre-wash	88	50	57%	1.17 ^c^
Head, post-wash	70	40	57%	0.61 ^def^
Course ground meat	67	46	69%	0.66 ^de^
Pork fat	40	28	70%	0.81 ^d^
Batter	66	42	64%	0.34 ^fg^
Final product	66	38	58%	0.43 ^efg^

^abcdefg^ Values in rows with no common superscripts differ (*p* < 0.05).

**Table 3 microorganisms-12-01599-t003:** Concentration, pre-enrichment prevalence, and post-enrichment prevalence of Salmonella from pork samples (*n* = 830) collected at various stages of sausage manufacturing.

	Pre-Enrichment Prevalence				Post-Enrichment Prevalence		
Stage	Samples	Positives	Prevalence	Log_10_ CFU/mL	Samples	Positives	Prevalence
Hide, pre-wash	88	64	73%	2.66 ^a^	88	83	94%
Hide, post-wash	88	1	1%	0.28 ^b^	88	23	26%
Carcass, pre-evisceration	88	10	11%	0.16 ^bcd^	88	42	48%
Carcass, post-trim	85	10	12%	0.23 ^bc^	85	30	35%
Carcass, post-wash	84	3	4%	0.09 ^cd^	84	10	12%
Head, pre-wash	88	3	3%	0.14 ^bcd^	88	17	19%
Head, post-wash	70	7	10%	0.02 ^d^	70	15	21%
Course ground meat	66	0	0%	0.02 ^d^	67	14	21%
Pork fat	40	0	0%	–	40	0	0%
Batter	66	0	0%	0.01 ^d^	66	10	15%
Final product	66	1	2%	0.02 ^d^	66	10	15%

^abcd^ Values in rows with no common superscripts differ (*p* < 0.05).

## Data Availability

Data available from authors upon request.

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
