# Peer review of "Prevalence and Characterization of Salmonella during Pork Sausage Manufacturing"

_microorganisms, 2024, doi:10.3390/microorganisms12081599_

Round 1

Reviewer 1 Report

Comments and Suggestions for Authors

The manuscript is very interesting, even if it is a brief report. The topic addressed is of interest as pork and its products contamination with pathogens is mandatory for consumers safety, given that pork meat is a beloved type of meat in many countries. In my opinion, the authors could collect more data and provide a full length article. 

Considering now the bried report is well structures and organized. The English language is good and the authors have discussed well their findings. I found some minor issues that must be addressed. I have indicated these within the attached pdf.

Based on my overall evaluation and the importance of this research I suggest a minor revision.

Sincerely,

Comments on the Quality of English Language

Author Response

Comment: remained (not: “remainder of”) (Abstract, line 26)

Response: “remainder of” is grammatically correct. Alternatively, “remaining” could be used here.

Comment: Why did you use linear mixed effect regression? Provide the hypothesis behind this selection.

Response: A linear mixed effects model was used in order to explore the count data and account for the dependencies between the variables in the study.  A linear model was chosen due to the continuous log10CFU data.  A mixed effects model was chosen in order to include both fixed and random effects.  Sample type was included as a fixed effect.  Month was included as a random effect to represent supplier.

Comment: Include a reference or references here.

Response: We have added references.

Reviewer 2 Report

Comments and Suggestions for Authors

This study of bacteria in pork carcasses and meat found that the interventions had beneficial effects to reduce the concentrations of Salmonella and Enterobacteriaceae. However, there are several problems that need to be solved in this paper:

1.The unit of ℃ in the text is wrong, and there is a space between the number in line 114. Please check such errors in the full text.

2.The abstract should focus on the background and content of the research.

3.Considering the time span, spatial span and diversity of these samples, how does the author guarantee the sampling accuracy?

4.Some bacterial strains in meat have developed resistance. Can interventions deal with these resistant strains?

5.The format of the references in this paper is not uniform and the years are too old.

6.Whether the sample identified as "zero" is not included will affect the accuracy of the entire experiment?

7.The results chapter lacks critical analysis and the analysis is insufficient.

8.The discussion chapter should not contain too much background content and should focus on the summary and prospect of the experiment.

Comments on the Quality of English Language

 Moderate editing of English language required

Author Response

Comment: This study of bacteria in pork carcasses and meat found that the interventions had beneficial effects to reduce the concentrations of Salmonella and Enterobacteriaceae. However, there are several problems that need to be solved in this paper:

Response: Thank you for the comments. We believe they have helped improve the paper. Comments are addressed individually below.

Comment 1: The unit of â„ƒ in the text is wrong, and there is a space between the number in line 114. Please check such errors in the full text.

Response: Checked and corrected.

Comment 2: The abstract should focus on the background and content of the research.

Response: The abstract has been modified.

Comment 3: Considering the time span, spatial span and diversity of these samples, how does the author guarantee the sampling accuracy?

Response: Because personnel from the manufacturing plant collected all samples and provided data on source farms, sample collections cannot be guaranteed to follow the specific protocol. We now include this potential source of variability in the discussion.

Comment 4: Some bacterial strains in meat have developed resistance. Can interventions deal with these resistant strains?

Response: We did conduct antimicrobial susceptibility testing. Further analysis of antimicrobial resistance was beyond the scope of this paper. Post-harvest treatment kills bacteria regardless of antimicrobial resistance profiles.

Comment 5: The format of the references in this paper is not uniform and the years are too old.

Response: Format has been corrected. We believe it is important to include older references that include original work when appropriate.

Comment 6: Whether the sample identified as "zero" is not included will affect the accuracy of the entire experiment?

Response: We, of course, don’t know what the post-enrichment prevalence would have been for Enterobacteriaceae but the pre-enrichment prevalence was already quite high. An additional post-enrichment step to show an even higher prevalence was not deemed to be helpful for Enterobacteriaceae. For Salmonella the post-enrichment was useful and was conducted.

Comment 7: The results chapter lacks critical analysis and the analysis is insufficient.

Response: We have added two charts that summarize the prevalence of Salmonella at the various stages.

Comment 8: The discussion chapter should not contain too much background content and should focus on the summary and prospect of the experiment.

Response: The discussion has been modified to more clearly reflect the data in this study.

Round 2

Reviewer 2 Report

Comments and Suggestions for Authors

accept